# Stretchy Electrochemical Harvesters for Binarized Self-Powered Strain Gauge-Based Static Motion Sensors

**DOI:** 10.3390/s22124542

**Published:** 2022-06-16

**Authors:** Hyeon Jun Sim, Jeeeun Kim, Jin Hyeong Choi, Myoungeun Oh, Changsoon Choi

**Affiliations:** Department of Energy and Materials Engineering, Dongguk University, Seoul 04620, Korea; shj0531@dongguk.edu (H.J.S.); cntjr456@dgu.ac.kr (J.K.); locomotive97@dongguk.edu (J.H.C.); abcdeun@dongguk.edu (M.O.)

**Keywords:** mechano-electrochemical energy harvester, self-powered strain sensor, stretchable, soft, wearable

## Abstract

The human monitoring system has motivated the search for new technology, leading to the development of a self-powered strain sensor. We report on the stretchable and soft stretchy electrochemical harvester (SECH) bilayer for a binarized self-powered strain gauge in dynamic and static motion. The active surface area participating in the electrochemical reaction was enhanced after stretching the SECH in the electrolyte, leading to an increase in the electrochemical double-layer capacitance. A change in the capacitance induced a change in the electrical potential of the bilayer, generating electrical energy. The SECH overcomes several challenges of the previous mechano-electrochemical harvester: The harvester had high elasticity (50%), which satisfied the required strain during human motion. The harvester was highly soft (modulus of 5.8 MPa), 103 times lower than that of the previous harvester. The SECH can be applied to a self-powered strain gauge, capable of measuring stationary deformation and low-speed motion. The SECH created a system to examine the configuration of the human body, as demonstrated by the human monitoring sensor from five independent SECH assembled on the hand. Furthermore, the sensing information was simplified through the binarized signal. It can be used to assess the hand configuration for hand signals and sign language.

## 1. Introduction

The human monitoring system has motivated the search for new technologies, such as artificial electronic skin, ubiquitous healthcare, organ motion tracking, soft robotics, and biomedical devices [1,2,3]. Developing such a system into a wearable device form requires several features—stretchability, softness, lightweight, continuous motion sensing in real-time, and simplicity—to prevent restricting human motion in daily life [4,5]. Previous research reported a wearable strain gauge, which measures the dynamic- and static-strain through resistance value [1,2,3,4,5]. However, the external electrical energy source was required to measure the resistance, which inevitably induces bulkiness and complexity.

Therefore, the mechanical energy harvester, a system that converts mechanical energy into electrical energy, is in demand to overcome these limitations, as a self-powered strain sensor that operates without an external energy source [6,7,8,9,10,11,12,13,14]. One study proposed the piezoelectric generator for a self-powered strain sensor. However, the sensitivity reduced enormously in low-frequency movement below 2 Hz, which is a limit as a sensor that measures the human motion between 0.1 Hz and 5 Hz [6,7,8]. The triboelectric generator shows high performance; it is considered useful for future wearable sensors [9,10,11,12,13,14]. However, it faces difficulties in measuring the static strain, due to instance current flow [13,14].

Recently, a mechano-electrochemical harvester—a system that induces the electrical potential gradient from mechanical stress by the electrochemical principle—has been reported for self-powered strain sensors [15,16,17,18]. The harvester was optimized for low-speed motion-sensing below 5 Hz, because it can generate a continuous current by ion diffusion kinetics, making it suitable for wearable sensors [18]. For instance, the multi-walled carbon nanotube (MWNT)/polyester thread-based bendable harvester shows a continuous electrical response to human motion below 1 Hz [15]. However, only a bending motion is measured, due to its relatively low strain. Alternatively, the coiled MWNT yarn demonstrates excellent performance as an energy harvester and wearable sensor during stretching [18]. However, its usage in practical applications is limited, due to the high cost of the MWNT sheet [19], the instability of the coil that tries to untwist by itself [20], high elastic modulus (up to 600 MPa) [18], and difficulty in scaling up from synchronization [21].

This paper presents the stretchable and soft stretchy electrochemical harvester (SECH) for a binarized self-powered strain gauge in dynamic and static motion. Our main strategy was to generate electrical energy owing to the change in the surface area of the carbon composite in the electrolyte. The MWNT was in direct contract with the electrolyte in the case of the MWNT/styrene-ethylene-butylene-styrene (SEBS) bilayer fabricated by spray-coating. The bilayer maintained its electrical conductivity during stretching (50%), due to the physical network entangled between the MWNTs. The active surface area participating in the electrochemical reaction was enhanced after it was stretched in the electrolyte, leading to an increase in the electrochemical double-layer capacitance (EDLC). Conversely, the stretched bilayer returned to its initial length due to the elastomer after the stretching force was removed, leading to a decrease in the capacitance. According to Q = CV [18], a change in the capacitance induced a change in the electrical potential of the bilayer, generating electrical energy from mechanical energy. The SECH overcame several challenges of the previous mechano-electrochemical harvester: (1) The SECH demonstrated high elasticity (50%). The human body requires 50% of elasticity in their joints during daily life [21]; however, the previous mechano-electrochemical harvester found it difficult to satisfy this elasticity (below 20%) [15,16,17,22]. The elasticity of the composite was achieved while having electrical conductivity by implementing a carbon/elastomer bilayer-based spray-coating; (2) The SECH is very soft (modulus of 5.8 MPa). The critical point in applying it as a wearable device is that it does not restrict human motion, while the previous harvesters acquired high stress under stretching due to their high modulus (up to 600 MPa) [18]. The modulus of the SECH is 103 times lower than that of the previous harvester, and it is suitable for the wearable device because it is similar to the modulus of the human body [23].

The SECH is applicable to a self-powered strain gauge, which is capable of measuring stationary deformation and low-speed motion. The SECH gauge created an electronic skin-sensing system to examine the configuration of the human body, as demonstrated by a human monitoring sensor from five independent gauges assembled on a hand. Highly soft, stretchable, and super-thin gauge minimizes the wearer’s discomfort in daily life. It can successfully measure the static strain, unlike conventional piezoelectric generators. Furthermore, the sensing information was simplified through the binarized signal and could be used to assess the hand configuration for hand signals and sign language.

## 2. Materials and Methods

The MWNT/SEBS bilayer was fabricated using commercial MWNT powder (Sigma-Aldrich, St. Louis, MO, USA) and SEBS (Sigma-Aldrich, St. Louis, MO, USA). Furthermore, 0.1 wt.% of the MWNT powder was dispersed in water with 1 wt.% sodium dodecylbenzene sulfonate (SDBS (Sigma-Aldrich, St. Louis, MO, USA)). We coated 10 mL dispersed solution with a spray gun on the Teflon film (20 cm × 20 cm) (Sigma-Aldrich, St. Louis, MO, USA); the coated film was dried in an oven at 60 °C for 3 h. After that, 4 wt.% of SEBS was dissolved in chloroform with a magnetic stirrer for 24 h. The dissolved solution of 10 mL was spray-coated on the MWNT layer, and this elastomer coating was repeated five times to achieve an even coating. After drying completely, the layer peeled off easily, due to its poor adhesion to the Teflon film. The electrochemical performance was measured using a electrochemical analyzer (Gamry instrument, Warminster, PA, USA, model G750). The electrochemical measurements used a three-electrode system comprising a working electrode, a counter electrode of platinum mesh, and an Ag/AgCl reference electrode. The energy production was measured with a SECH connected to an external resistance to measure power and electrical energy. The voltage applied to the external resistor was measured with an oscilloscope by changing the external resistance (Tektronix, Beaverton, OR, USA, DPO4014B). The power (P = V^2^/R, where P denotes power, V represents voltage, and R is resistance) was measured using the results obtained from the oscilloscope. A field emission scanning electron microscope (FESEM, Hitachi S4700, Tokyo, Japan) (at 15 kV) was used to observe the morphology. Furthermore, the electrical measurements were performed using a digital multimeter (Model 187, Fluke Corporation, Everett, WA, USA). The mechanical test was performed using a universal tester (UTM, INSTRON 5966, INSTRON, Norwood, MA, USA). A SECH was cut in the form of a strap (0.5 cm × 2 cm) to provide a self-powered strain sensor for the artificial e-skin system. After cutting it and attaching it to the finger, both ends were fixed with carbon tape, and the electric wire was connected. The potential and current were measured in saline with a electrochemical analyzer.

## 3. Result and Discussion

### 3.1. The Morphology and Characteristics of Multi-Walled Carbon Nanotube/Styrene-Ethylene-Butylene-Styrene Bilayer

The commercial MWNT, dispersed in water and the SEBS solution, was sequentially spray-coated, resulting in a MWNT/SEBS bilayer (Figure 1a and Appendix A). The spray-coating is a fascinating manufacturing method for mass production and industrialization, because it is easy to take it to a large scale through a continuous process. This bilayer can produce a wide form with a width, length, and thickness of 20 cm, 20 cm, and 30 µm (Appendix A). The spray-coating is also a powerful tool to print the desired shape using a mask, allowing for serpentine printing with a 2 mm radius (Figure 1b and Appendix A). The bilayer had elasticity and softness along with electrical conductivity due to the bilayer structure (Figure 1c). Contrary to the previous homogeneous composite, the bilayer had a relatively high electrical conductivity (3.5 KΩ/sq) despite the low weight percent (wt.%) of MWNT (below 5 wt.%). The MWNTs formed a physical network with each other after the conductive MWNT and the elastic SEBS layers were separated through the sequential coating (Figure 1d). The mechanical entanglement by the Van der Waals force formed the MWNT membrane, contributing to the electrical pathway (Appendix A) [24]. Additionally, the MWNT layers had a high active surface area for the electrochemical reaction, because they were directly exposed to the surface (Figure 1e). Furthermore, the polymer layer maintained its elasticity and softness without hardening by the MWNT. It avoided the wearer’s discomfort while in contact with the human skin due to the softness, elasticity, and micro-scale thickness of the bilayer. The density of the bilayer was significantly light (2.7 mg/cm^2^ and 0.9 g/cm^3^), approximately similar to the density of SEBS, because of the high wt.% of the SEBS in the bilayer.

Furthermore, the bilayer demonstrated stretchable and soft electrode performance. The bilayer could be reversibly stretched at 50% strain, 2.9 MPa stress (Figure 1f). Humans require a high degree of strain in daily life. For instance, the joints, such as the wrist, knee, and elbow, experience more than 50% strain [21]. The bilayer is suitable for the wearable device because its high elasticity minimizes the movement restrictions of the wearer. Moreover, softness is a critical issue for a wearable device, for minimizing the discomfort of the wearer. The high modulus (up to 600 MPa) of the previous stretchable mechano-electrochemical harvester [18], which is higher than that of human muscle (10 MPa) [23], may cause problems, such as damage to the body and inconvenience for the wearer. Our bilayer showed a low modulus (5.8 MPa), 103 times lower than that of the previous harvester, making it suitable for a wearable device. The bilayer maintained its conductivity function during the mechanical deformation (Figure 1g). A MWNT/SEBS strap (0.5 cm × 4 cm × 30 µm) was fixed to a Vernier caliper, and the resistance value according to the strain change was measured. When the strain increased to 50%, the resistance ratio divided by the initial resistance increased to 2.9. The resistance ratio returned to its initial value because its length returned to its original length. The function of the stretchable conductor originated from the internal MWNT network. When the bilayer was stretched, the weakening and detachment of the MWNT contact resulted in a smaller overall contact, increasing the electrical pathway. When the bilayer was released, the MWNT slid back by a certain degree with the polymer. Therefore, even when the strain was repeatedly applied, the resistance of the bilayer had a specific value depending on the strain.

### 3.2. Mechanism of Stretchy Electrochemical Harvesting Bilayer from Surface Area Change in Electrolyte

When the bilayer was stretched in the electrolyte, the electrical energy was generated from the mechanical stimulation of a SECH. The electrochemical cell was set up to verify the performance of the energy harvester (Figure 2a and Appendix A). The cell was composed of a working electrode of MWNT/SEBS bilayer, the counter electrode of platinum (Pt) mesh, and the reference electrode of Ag/AgCl. All of the components were immersed in the aqueous electrolyte of saline for the biomedical system. When the bilayer was immersed in the electrolyte, a chemical potential difference existed between the electrode surface and surrounding electrolyte [18,25]. Simultaneously, the MWNTs formed an EDLC with the ion of the electrolyte. The active surface area participating in the electrochemical reaction was enhanced because the bilayer was stretched to 50% (Figure 2b), increasing the capacitance by 6.8% from 11.1 mF/cm^2^ to 11.8 mF/cm^2^ (Figure 2d).

According to piezoelectrochemical spectroscopy (PECS) analysis [18], the potential of the zero charge (PZC) value of the bilayer in saline was 650 mV compared to that in the Ag/AgCl (Figure 2c). The cation was expected to act mainly on the surface because the potential value (180.2 mV versus Ag/AgCl) was relatively lower than that of PZC [25]. According to Q = CV (Q denotes the charge on the surface, C is the capacitance of the bilayer, V represents the intrinsic voltage), an increase in the capacitance with stretching causes a decrease in the absolute value of the intrinsic voltage (OCV-PZC) in open-circuit [18]. Hence, the open-circuit potential versus Ag/AgCl increased from 180.2 mV to 181.6 mV, when the bilayer was stretched by approximately 50% at a frequency of 1 Hz (Figure 2e). Additionally, when the bilayer was recovered to the initial state, the potential was recovered to 180.2 mV. The electrical current was continuously generated by ion movement through the mechanical deformation. Upon return to the releasing state, the active surface area decreased, and the reverse current re-established the equilibrium electrochemical potential (Appendix A). A continuous current was produced at low-frequency motion (1 Hz), which is better synchronized to the time scales of human and organ motion (from 0.1 Hz to 5 Hz) as a wearable self-powered strain sensor compared to alternative methods.

### 3.3. Characterization of the Stretchy Electrochemical Harvester in Slow Dynamic Movement with Various Conditions

The open-circuit voltage and short-circuit current of the SECH were measured under various conditions that change in the daily life of a human (Figure 3a). The electrochemical cell was set up to confirm the performance, which was composed of a working electrode of MWNT/SEBS bilayer and a counter electrode of Pt. mesh in a two-electrode system. The SECH can be easily scaled up by increasing the generated current according to the area. The various straps with a 0.5 cm width and lengths of 1 cm, 2 cm, and 4 cm were produced by cutting a large bilayer area. The short-circuit current was measured during the 1-Hz sinusoidal stretch to 50% strain in saline. The current increased from 0.14 µA to 0.93 µA as the area increased from 0.5 cm^2^ to 2 cm^2^ (Figure 3b). In the previous mechano-electrochemical harvester, several yarns were connected in parallel or series to amplify the output. However, the summed output in the sinusoidal output may be degraded, due to destructive interference caused by the synchronization [26]. All of the harvesters must move in the same phase to amplify the output through constructive interference; however, the human movement varies with position. The SECH, initially formed with a large area, solved the synchronization problem, because the force was evenly distributed by the elastomer.

We analyzed the voltage generation performance of the SECH, according to the applied strain (Figure 3c). The open-circuit voltage was measured during a 1-Hz sinusoidal stretch in saline. The increment of the applied strain on the bilayer from 10% to 50% increased the peak-to-peak voltage from 4 mV to 8 mV. Furthermore, the performance remained approximately constant with the frequency, electrolyte concentration, and temperature (Figure 3d and Appendix A). The generated electrical energy with external load is shown in Figure 3e. The peak-to-peak voltage and power were acquired with load resistance during the 1-Hz sinusoidal stretch to 50% strain in saline. The peak-to-peak voltage was increased with external load resistance, and the peak power of 156 µW/kg (standard deviation value of 13.27 with five different SECH samples) was maximized at a load resistance of 200 Ω (Figure 3e and Appendix A). At this time, the maximum area peak power is 1.02 mW/m^2^, which is similar to the performance of the existing film-type electrochemical energy harvester (1.28 mW/m^2^) [17]. The existing piezoelectric and triboelectric nanogenerators have high impedance matching values (above 100 MΩ) [10,11], making it difficult to integrate them into an electric circuit with another electric device. However, our device demonstrated a significant reduction in the internal resistance value to provide a high advantage as an electronic device. The performance stability during the repeated stretching/releasing cycle was significant. Figure 3f shows that the peak power was maintained for more than 1000 cycles during the 1-Hz sinusoidal stretch to 40% strain in saline. The open-circuit voltage remained approximately constant at 6 mV during the 1000 cycles; moreover, the voltage retention was changed below 3% (Appendix A).

### 3.4. Binarized Self-Powered Strain Gauge for Dynamic and Static Human-Motion Monitoring System

The SECH can be applied to a self-powered strain gauge, which can measure stationary deformation and low-speed motion. The integration of the SECH creates an electronic skin sensing system to examine the configuration of the human body, as demonstrated by the human monitoring sensor from five independent SECHs assembled on the hand (Figure 4a). The highly soft, stretchable, and very thin SECH was attached to the body in the form of electronic skin, minimizing the wearer’s discomfort in daily life. Most importantly, this SECH could successfully measure the static strain, unlike conventional self-powered strain sensors. Hence, the same-sized (0.5 cm × 2 cm) SECH and the commercially available piezoelectric film (silver-coated PVDF-TrFE, Piezotech, Pierre-Bénite, France) for the self-powered strain sensor were compared to confirm it. When the straps were attached to the finger joint, the open-circuit voltage and short-circuit current were measured while bending the finger. As the finger was bent, the SECH attached to the finger was stretched and this induced strain is the main reason for the electricity generation of our SECH. However, it is difficult to measure the static strain of the piezoelectric film using an open-circuit voltage or short-circuit current because the output peak momentarily only appears during the movement. In contrast, the open-circuit voltage was increased from −338 mV to −336 mV when the SECH was stretched (Figure 4b); it remained at −336 mV while the finger was held flexed (Figure 4c). Additionally, the short-circuit current showed a continuous output when the fiber was bent, similar to the voltage output (Figure 4d). The static strain can be measured through the formed potential even in a static state, because the electrical potential of the SECH is induced by ion dynamics adsorbed on the carbon surface, enabling its application as a self-powered strain gauge. The SECH gauge is lighter and simpler than the conventional optical and metal strain gauge system and does not restrict the range of motion of the hand [2]. The system can be used as a master hand to control remote slave robots either to perform surgical procedures remotely or to increase the safety and speed of mine clearance.

Furthermore, we propose a binary sensing system that can monitor the human-body configuration with only a small amount of data. An operator capable of extracting, transforming, and loading data and additional energy sources were required to drive the system to process the real-time wearable sensor’s big data. The increment in the amount of processed data increases the complexity and bulkiness of the system. Therefore, we reduced the sensing information by binarizing the output of the SECH, using the unique characteristic of continuously generating voltage in real-time with static strain. It is possible to distinguish the hand gesture by matching the binary number and the hand configuration. When the configuration of the hand changed according to the hand signal for help, the gesture can be reconstructed through the continuously responding binary signal of each finger (Figure 4e and Appendix A). Additionally, the gesture could be detected through the signal when the letters “H”, “E”, “L” and “P” were expressed based on the sign language, corresponding to the expression of the letters (Figure 4f). The electronic skin has significant potential in low-power transmission/reception systems for the ubiquitous healthcare system, smart robotics, and human–machine interaction.

## 4. Conclusions

A novel type of stretchable and soft stretchy electrochemical harvester can be used for binarized self-powered strain gauges, which can measure the static strain and slow motion without an external power source. The major novelty of this paper is in the first reported stretchy electrochemical harvesting bilayer through surface area change. Our main strategy was to generate electrical energy due to the change in the surface area of carbon composites in the electrolyte. When the SECH was stretched in the electrolyte, the active surface area participating in the electrochemical reaction was enhanced, increasing the electrochemical double-layer capacitance. A change in the capacitance induces a change in the electrical potential of the bilayer according to Q = CV, generating electrical energy through mechanical energy. The bilayer structure, in which the carbon electrode has the function of a stretchable electrode while in direct contact with the electrolyte, was designed to develop this harvester. In addition, this study shows the advanced characteristics of a SECH, that are more suitable for wearable sensors than before. (1) The SECH had high elasticity (50%). The human body requires up to 50% or more elasticity in the joints during daily life; however, the previous mechano-electrochemical harvester found it difficult to satisfy this elasticity (below 20%). The elasticity of the elastomer was maintained while still having electrical conductivity by implementing a carbon/elastomer bilayer composite-based spray coating; (2) The SECH was highly soft (modulus of 5.8 MPa). When applied as a wearable device, the device must be deformed with low stress in order to not restrict the human motion, whereas, the previous harvester requires high stress in stretching due to their high modulus (up to 600 MPa). The modulus of the SECH was 103 times lower than that of the previous harvester. Moreover, it is suitable as a wearable device, because it is similar to the modulus of the human body.

In addition, new applications of binarized self-powered strain gauges in dynamic and static motion open up new possibilities. The SECH can be applicable to the self-powered strain gauge, which is capable of measuring stationary deformation and low-speed motion. The SECH gauge created a system to examine the configuration of the human body, as demonstrated in the human monitoring system from five independent SECHs assembled on the hand. Although the highly soft, stretchable, and very thin SECH minimizes the wearer’s discomfort in daily life, it can successfully measure the static strain, unlike conventional piezoelectric generators. Furthermore, the sensing information was simplified through the binarized signal and could be used to assess the hand configuration for hand signals and sign language.

However, the low power density and environmental limitation, which was driven only in the electrolyte, still remain. We plan to conduct an experiment using bigger capacitance change and solid electrolytes in the future to solve this problem.

## Figures and Tables

**Figure 1 sensors-22-04542-f001:**
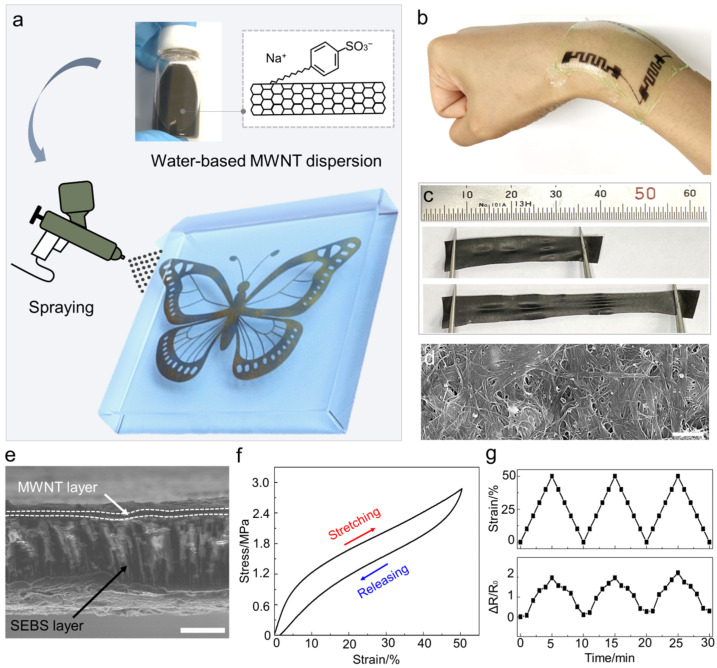
The morphology and characteristics of MWNT/SEBS bilayer-based electrochemical harvester. (**a**) Schematic illustration of fabrication process based on the sequential spray-coating method. A MWNT/SEBS bilayer was formed by sequentially coating MWNT dispersion and SEBS solution. (Inset) The optical image of MWNT dispersion with sodium dodecylbenzene sulfonate; (**b**) The optical image of the patterned MWTN/SEBS bilayer was attached to the human wrist to demonstrate the utility of the wearable device; (**c**) The optical image of MWNT/SEBS bilayer for 0% and 50% strain; (**d**) SEM image of MWNT/SEBS bilayer in which MWNT are physically entangled with each other (scale bar: 500 nm); (**e**) The cross-sectional SEM image of MWNT/SEBS bilayer (scale bar: 10 um); (**f**) Strain–stress curve of MWNT/SEBS bilayer stretched to 50% strain; (**g**) The strain and resistance ratio value of the MWNT/SEBS bilayer while reversibly stretching to 50% strain.

**Figure 2 sensors-22-04542-f002:**
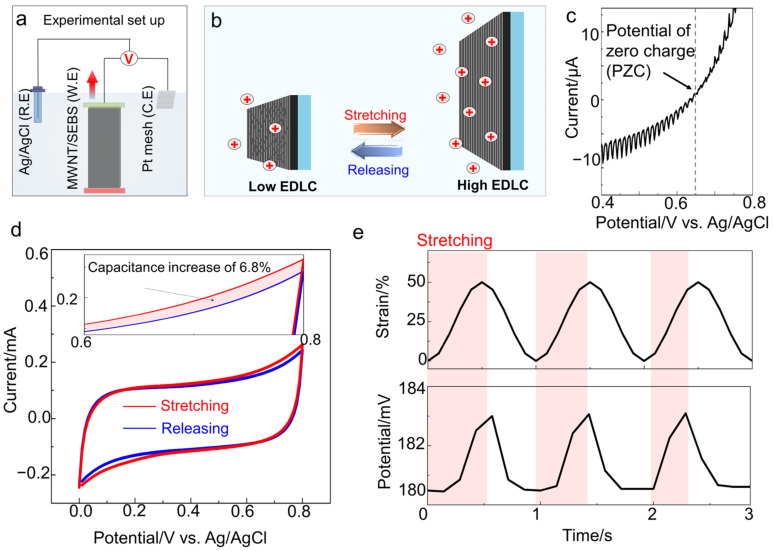
Mechanism of stretchy electrochemical harvester from surface area change in the electrolyte. (**a**) Schematic illustration of electrochemical experiment setting composed of working electrode of MWNT/SEBS bilayer, the counter electrode of Pt, and the reference electrode of Ag/AgCl; (**b**) Schematic illustration of stretchy electrochemical energy harvesting mechanism. The active surface area of MWNT changed with stretching and releasing state; (**c**) Piezoelectrochemical spectroscopy (PECS) analysis results of MWNT/SEBS bilayer in saline solution; (**d**) Cyclic voltammetry curve for 0% (blue line) and 50% strain (red line); (**e**) Sinusoidal applied tensile strain and resulting potential (vs. Ag/AgCl) during the 1-Hz sinusoidal stretch. The red area is in a stretched state.

**Figure 3 sensors-22-04542-f003:**
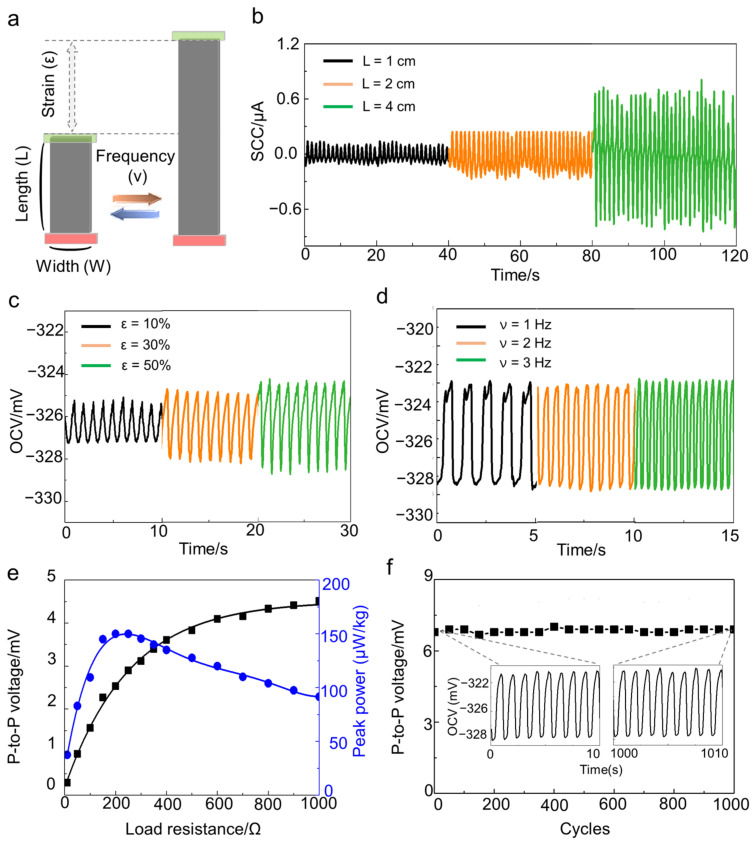
The characterization of the stretchy electrochemical harvester in slow dynamic movement with various conditions. (**a**) The schematic illustration of the harvester with various experimental conditions; (**b**) The short-circuit current output under only length increment from 1 cm to 4 cm at a constant width of 0.5 cm; The open-circuit voltage (**c**) with various applied strains from 10% to 50% during the 1-Hz sinusoidal stretch and (**d**) with various frequencies from 1 Hz to 3 Hz at a strain of 50% in saline; (**e**) The peak-to-peak voltage and peak power with external load resistance; (**f**) The stability of open-circuit voltage output during 1000 cycles in saline, (inset) the open-circuit voltage after first cycles, and 1000 cycles.

**Figure 4 sensors-22-04542-f004:**
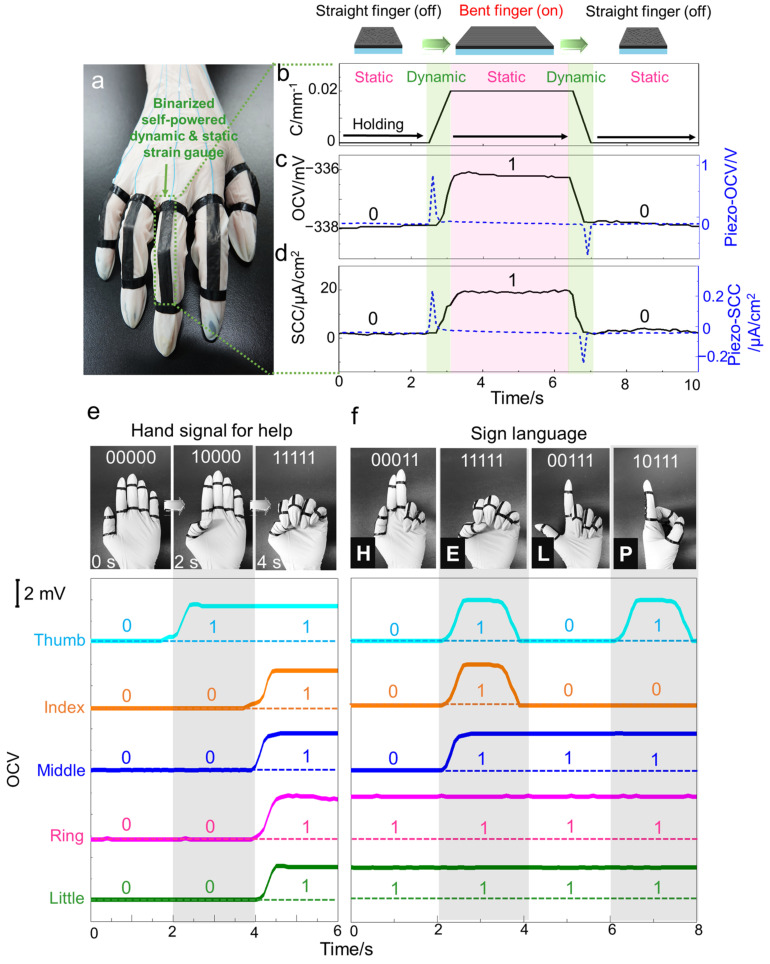
The binarized self-powered strain gauge for the dynamic and static human-motion monitoring system. (**a**) The optical image of wearable self-powered strain gauge using SECH; (**b**) The curvature; (**c**) open-circuit voltage; and (**d**) short-circuit current of SECH and commercial piezoelectric film when the finger was bent. The open-circuit voltage versus time with change of hand configuration for (**e**) hand signal for help and (**f**) sign language. (Inset) the photograph of hand configuration with a binary number.

## Data Availability

Not applicable.

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
