# Peer review of "Stretchy Electrochemical Harvesters for Binarized Self-Powered Strain Gauge-Based Static Motion Sensors"

_sensors, 2022, doi:10.3390/s22124542_

Round 1

Reviewer 1 Report

This paper proposes on the stretchable and soft stretchy-electrochemical harvester bilayer for binarized self-powered strain gauge in dynamic and static motion. These factors are also neglected in former works. And the results in this paper can make contribute to its engineering application. The paper can be considered for publication if the issues given in the following issues are discussed further:

(1) The article mainly analyzes the output voltage and current of the energy harvester. If it is to be used in a wearable health monitoring system for the human body, it is recommended that the author analyze the output power of the energy harvester.

(2) In the introduction part, the author's introduction to the energy harvester is relatively simple, and it is recommended to add some introductions of other energy harvesters.

(3) The readability and presentation of the study should be further improved. The paper suffers from language problems. The paper should be proofread by a native speaker or a proofreading agent.

Author Response

Response to Reviewer’s Comment

We appreciate the comments of the reviewers and the suggestions of the editor. Our responses to these comments are listed below, and the manuscript and supplemental materials have been accordingly revised.

(1) The article mainly analyzes the output voltage and current of the energy harvester. If it is to be used in a wearable health monitoring system for the human body, it is recommended that the author analyze the output power of the energy harvester

Answer: Thank you for your comments and suggestions. The Power (P = V2/R, where P denotes power, V represents voltage, and R is resistance) was measured using the result obtained from the oscilloscope. The peak-to-peak voltage was increased with external load resistance, and the maximum peak power of 156 µW/kg was measured at load resistance of 200 Ω (Fig. 3(e)).

The explanation was added in the revised manuscript (line 238-240)

(2) In the introduction part, the author's introduction to the energy harvester is relatively simple, and it is recommended to add some introductions of other energy harvesters.

Answer: According to your comment, some introduction of other energy harvesters was added in the revised manuscript.

“ Therefore, the mechanical energy harvester, a system that converts mechanical energy into electrical energy, is in demand to overcome these limitations as a self-powered strain sensor that operates without an external energy source [6–14]. One study proposed the piezoelectric generator for a self-powered strain sensor. Piezoelectric device based on the aligned electors spun mats has high sensitivity for measuring small pressure [6] and core-shell piezoelectric film measured the bending motion on a human arm [8]. However, the sensitivity reduced enormously in low-frequency movement below 2 Hz, which is a limit as a sensor that measures the human motion between 0.1 Hz and 5 Hz [6–8]. Triboelectric generator shows high performance; it is considered useful for future wearable sensors [9–14]. Fiber and textile-based triboelectric nanogenerator sense the different kind of body motion (elbow, knee, stepping and tapping motion)[9]. However, it faces difficulties in measuring the static strain due to instance current flow [13,14].

Recently, mechano-electrochemical harvester—a system that induces the electrical potential gradient from the mechanical stress by the electrochemical principle—has been reported for self-powered strain sensors [15–18]. The harvester was optimized for low-speed motion-sensing below 5 Hz because it can generate continuous current by ion diffusion kinetics, making it suitable for wearable sensors [18]. For instance, the multi-walled carbon nanotube (MWNT)/polyester thread-based bendable harvester shows the continuous electrical response to human motion below 1 Hz [15].“

The explanation was added in the revised manuscript (line 37-55)

(3) The readability and presentation of the study should be further improved. The paper suffers from language problems. The paper should be proofread by a native speaker or a proofreading agent.

Answer: The revised manuscript was proofread by a proofreading agent.

Reviewer 2 Report

Journal Name: Sensors

Title: Stretchy-electrochemical harvesters for binarized self-powered strain gauge based static motion sensors

It’s an interesting and suitable article for the sensors journal. The article fits the scope of the journal. I recommend its publication with minor revision. Here I am mentioning my detailed comments.

11.      Line number 66 has an equation; although it is a basic equation reference will be useful for nonelectrochemical science students

22.      Can a single-walled carbon nanotube be useful for this purpose.

33.      The size of the Figure 2c-e is very low and the size has to be enhanced

44.      What is the reason for the minute enhancement of OCV when SECH was bent? Has to be explained

55.      Conclusion is too short and has to be improved

66.      Can the result be reproducible? need to be explained 

Author Response

Response to Reviewer’s Comment

We appreciate the comments of the reviewers and the suggestions of the editor. Our responses to these comments are listed below, and the manuscript and supplemental materials have been accordingly revised.

  1. Line number 66 has an equation; although it is a basic equation reference will be useful for nonelectrochemical science students

Answer: The reference related to the equation was cited.

“According to Q = CV[18], a change in the capacitance induced a change in the electrical potential of the bilayer, generating electrical energy from mechanical energy.”

The explanation was corrected in the revised manuscript (line 66-67)

  1. Can a single-walled carbon nanotube be useful for this purpose.

Answer: It is expected that single-walled carbon nanotube incorporated SECH show improved performance due to higher electrical conductivity than multi-walled carbon nanotube. We have a plan to conduct experiments for single-walled carbon nanotube based device in further work in the future.

  1. The size of the Figure 2c-e is very low and the size has to be enhanced

Answer: According to your comment, the size of figure 2c-e was enhanced.

  1. What is the reason for the minute enhancement of OCV when SECH was bent? Has to be explained

Answer: As the finger was bent, the SECH attached to the finger was stretched and this induced strain is the main reason for electricity generation of our SECH. Therefore, the open-circuit voltage was increased from -338 mV to -336 mV when the SECH was stretched (Fig. 4(b)).

The explanation was added in the revised manuscript (line 264-266)

  1. Conclusion is too short and has to be improved

Answer: Conclusion was improved.

“A novel type of stretchable and soft stretchy-electrochemical harvester can be used for binarized self-powered strain gauges, which can measure the static strain and slow motion without an external power source. The major novelty of this paper is first reported stretchy-electrochemical harvesting bilayer through surface area change. Our main strategy was to generate the electrical energy due to change in the surface area of carbon composite in the electrolyte. When SECH was stretched in the electrolyte, the active surface area participating in the electrochemical reaction was enhanced, increasing the electrochemical double-layer capacitance. Change in the capacitance induces a change in the electrical potential of the bilayer according to Q = CV, generating electrical energy through mechanical energy. The bilayer structure, in which the carbon electrode has the function of the stretchable electrode while in direct contact with the electrolyte, was designed to develop this harvester.

Also, this study has advanced characteristics suitable for wearable sensors than before. (1) The SECH had high elasticity (50%). The human body requires up to 50% or more elasticity in the joint during daily life; however, the previous mechano-electrochemical harvester found it difficult to satisfy this elasticity (below 20%). The elasticity of elastomer was maintained while having electrical conductivity by implementing a carbon/elastomer bilayer composite-based spray coating. (2) The SECH was highly soft (modulus of 5.8 MPa). When applied as a wearable device, the device must be deformed with low stress in order to not to restrict human motion, whereas, the previous harvester requires high stress in stretching due to their high modulus (up to 600 MPa). The modulus of SECH was 103 times lower than that of the previous harvester. Moreover, it is suitable for the wearable device because it is similar to the modulus of the human body.

In addition, new applications of binarized self-powered strain gages in dynamic and static motion open up new possibilities. The SECH can be applicable to the self-powered strain gauge, which is capable of measuring stationary deformation and low-speed motion. The SECH gauge created a system to examine the configuration of the human body, as demonstrated in the human monitoring system from five independent SECH assembled on the hand. Although highly soft, stretchable, and very thin SECH minimizes the wearer’s discomfort in daily life, it can successfully measure the static strain, unlike conventional piezoelectric generators. Furthermore, the sensing information was simplified through the binarized signal and could be used to assess the hand configuration for hand signals and sign language.

However, the low power density and environmental limitation, which was driven only in the electrolyte, still remain. We plan to conduct an experiment using bigger capacitance change using single-walled carbon nanotube and solid electrolytes in the future to solve this problem.”

The explanation was added in the revised manuscript (line 302-336)

  1. Can the result be reproducible? need to be explained

Answer: The reproducibility of the present work can be inspected by measuring peak power of the SECH during repeated strain application. As a result, the average peak power of the five different SECH samples is 156 µW/kg against 50% strain and their standard deviation value is 13.27, showing the reproducible performance.

Reviewer 3 Report

Dear Authors, please consider my comments in the attached pdf.

Author Response

Response to Reviewer’s Comment

We appreciate the comments of the reviewers and the suggestions of the editor. Our responses to these comments are listed below, and the manuscript and supplemental materials have been accordingly revised.

  1. if more than 50% is needed, what is your schedule to improve the SECH characterisics (limited for 50%)

Answer: The limit elasticity of SECH can be improved by adopting the micro-wrinkle structure on the CNT active layer. Generally, strain mismatch between different layers can form the wrinkle as previously reported literature (Changsoon Choi et al., Adv. Energy Mater. 2016, 1602021, Changsoon Choi et al., Nano lett., 2016, 16, 7677). In our work, based on the literatures, CNT spray coating on the pre-stretched elastomer substrate is expected to result in wrinkle formation after the strain removal. We have a plan to conduct the experiments for wrinkle structure based device in further work in the future. 

  1. I don’t know a gamry instrument. What does it do.

Answer: Gamry instrument is electrochemical analyzer that perform galvanostat/potentiostat test, which is used to measured the open-circuit voltage and CV curve for our SECH. The explanation was added in the revised manuscript (line 95-97)

  1. All quantities in diagrams should be give as e.g stress/MPa or strain/%, R/R0, time/min

Answer: According to your comment, all quantities in diagrams was corrected in the revised manuscript.

  1. please compare to literature results of similar devices.

Answer:  At this time, the maximum area peak power is 1.02 mW/m2, which is similar to the performance of the existing film-type electrochemical energy harvester (1.28 mW/m2) [17].

The explanation was added in the revised manuscript (line 241-242)